Recently, physics-informed neural networks (PINNs) have offered a powerful new paradigm for solving forward and inverse problems relating to differential equations. Whilst promising, a key limitation to date is that PINNs struggle to accurately solve problems with large domains and/or multi-scale solutions, which is crucial for their real-world application. In this work we propose a new approach called finite basis physics-informed neural networks (FBPINNs). FBPINNs combine PINNs with domain decomposition and separate subdomain normalisation to address the issues related to scaling PINNs to large domains, namely the increasing complexity of the underlying optimisation problem and the spectral bias of neural networks. Our experiments show that FBPINNs are more effective than PINNs in solving problems with large domains and/or multi-scale solutions, potentially paving the way to the application of PINNs on large, real-world problems.
