# OpenReview forum: "Scaling physics-informed neural networks to large domains by using domain decomposition"
_NeurIPS.cc/2021/Workshop/DLDE — DLDE Workshop -- NeurIPS 2021 Poster_

### Official Review · Reviewer_zvk1 · 2021-10-11

**Confidence:** 3

**Review:**

The authors present a methodology for applying physics-informed neural networks (PINNs) to large domains while mitigating the scalability and spectral bias problems inherent of said networks. They divide the domain in overlapping subdomains and normalise the input variables within each subdomain. A NN is trained for each subdomain.

The authors test their method over differential equations and show through these examples that their method outperforms the PINNs. They also highlight some limitations of the methodology: it can be worth to fine-tune the division of the subdomain in each particular problem.

The paper is clear and well written.

Comments:
-	The operator $\mathcal{C}$ of equation (3) is introduced in line 56 but not described in the paper (the authors reference the reader to [1]). However, as this operator allows to automatically satisfy the boundary conditions (line 56), I would suggest incorporating a paragraph describing it and its properties.
-	While describing the methodology, the authors indicate that $unnorm$ is a "common unnormalization" unique for all subdomains (line 53). I would recommend commenting on how this function should be selected. In particular, they could describe how it was selected for each of the given examples.
-	Figure 2 (a) and (e) suggest that the number of subdomains (30) has been selected to be equal to $\frac{1}{2\pi}\omega\Delta x$ where $\omega$ is the highest frequency and $\Delta x$ is the length of the domain. Is this a coincidence? Does the method also work for another decomposition scheme (e.g. using 29 or 31 subdomains)? The choice of the domain decomposition should be better motivated.
-	The problems presented in the Results section might be considered rather simple since their analytic solutions are easy to calculate. I would suggest incorporating some more complex differential equations.


**Score:**

2: Borderline paper

---

### Official Review · Reviewer_yKG1 · 2021-10-11

**Confidence:** 4

**Review:**


The authors propose to use PINN, but separating the domain into windows and then normalizing and running a different neural network in each window.

Learning higher frequencies much more slowly than lower frequencies is not major problem, it shouldn't prevent learning the correct solution given enough training time. It hasn't been a major issue in other applications, so why would it be in this one? So the premise that we should cut the problem into separate window seems wrong. One should only need a good enough neural network architecture, having to use 30 different neural networks to solve a toy 1D problem seems crazy. I can't imagine how much networks would be needed in large domains. If R^p, would we use 15^p different neural networks? This is not scalable.

It would be important to show the total number of parameters for the FBPINN vs the PINN to really show that its more parsimonious in number of parameters. Ideally you would compare in a setting with equal number of parameters at various size. Ex : 30 neural-nets with 500 parameters vs 1 neural-net with 500 * 30 parameters, 30 neural-nets with 1000 parameters vs 1 neural-net with 1000 * 30 parameters, etc. This could be a plot with one line/curve for each number of neural networks (e.g., 100, 500, 1000, 2500) and the x-axis would be the total number of parameters. This would really help convince that this idea is worthwhile.

Also optimizing the architecture a bit would be good to help convince that FBPINN is really better, because in my view, a bigger network with the right activations, layer-normalization, structure (residual-layer) would do better or as well as the multitude of small neural networks in small windows. Keeping to simple toy datasets is okay, but more experiments at equal computation are needed to convince. And you need at least one large dimension experiment, like one in R^100 and again show that FBPINN is better with the same number of total parameters than PINN.

The fact that FBPINN did not work on the Burgers equation due to discontinuity is again a sign to me that we shouldn't separate the domain into windows.

I don't believe in the premise that the domain should be cut into windows and the experiments are not convincing these doubts.

**Score:**

1: Reject: trivial or wrong

---

### Official Review · Reviewer_oTGg · 2021-10-12

**Confidence:** 3

**Review:**

This paper is well written. It proposes using multiple neural networks, one for each subdomain, instead of a single PINN over the entire domain. Because increasing domain size causes low frequencies in the solution to be viewed as high frequency features by a neural network, the argument is that using individual neural networks will more effectively learn high frequencies due to the smaller domains seen by each neural network, compared to a single PINN over the entire domain.

This approach of dividing the domain seems straightforward and reasonable, although not novel. The value of this work can be improved by comparing the tradeoffs in performance, compute, and memory between one PINN and the multiple neural networks in a FBPINN. Also, there is the question of whether this is scalable, given the increase in the number of neural networks. The authors can also better describe the procedure for designing and applying a good FBPINN. For example: “Any domain decomposition can be used, as long as the 54 subdomains overlap” → How should the domain decomposition be selected? How to compare and evaluate a particular domain decomposition against alternatives?

Figure 2 b), d), f), i) show that both FBPINN and PINN perform similarly against the exact solution, so it is unclear what the argument is for using FBPINN over PINN from those four subfigures. Figures g) and h) and the text describe faster convergence of FBPINN, but the differences can be highlighted in subfigures b), d), f), i). Figure 3 e) and f) are more useful.


**Score:**

2: Borderline paper

---

### Decision · Program_Chairs · 2021-10-16

**Decision:**

Accept (Poster)

**Comment:**

Reviewers generally seemed to be skeptical about this paper.

One significant concern raised was the scalability of the grid-like spatial decomposition, which increases exponentially with the dimension. Indeed PINNs have so far achieved the most success in "complicated" regimes such as high-dimensional PDEs, for which traditional (often grid-based) methods are lacklustre. In low-dimensional regimes they are usually often outperformed by traditional solvers. As such it seems that a hybrid approach may produce a method that is not competitive in any regime. The lack of evidence to the contrary is definitely the single greatest weakness of the paper.

On a more positive note, reviewers agreed that the paper was clearly written. I commend the authors for a paper that is far above the usual standard in this regard.

I do believe the principle of the proposed divide-and-conquer approach to be a reasonable one. As a practical matter, it is no doubt much easier to fit each small subnetwork to a small problem, than it is to fit a single large network to the whole thing.

Overall I am inclined to accept the paper. The proposed approach has clear limitations, but fixing them seems like it would make for an interesting line of work, and this paper is of the discussion-provoking kind that is a good fit for a workshop.